# Novel Design of Neuropeptide-Based Drugs with β-Sheet Breaking Potential in Amyloid-Beta Cascade: Molecular and Structural Deciphers

**DOI:** 10.3390/ijms23052857

**Published:** 2022-03-05

**Authors:** Cosmin Stefan Mocanu, Marius Niculaua, Gheorghita Zbancioc, Violeta Mangalagiu, Gabi Drochioiu

**Affiliations:** 1Faculty of Chemistry, Alexandru Ioan Cuza University of Iasi, 11 Carol I Bd., 700506 Iasi, Romania; cosmin.stefan.m@gmail.com (C.S.M.); gheorghita.zbancioc@uaic.ro (G.Z.); 2Research Centre for Oenology Iași, Romanian Academy Iași Branch, 8 Carol I, 700505 Iasi, Romania; niculaua@gmail.com; 3Department of Exact and Natural Sciences–CERNESIM Center, Institute of Interdisciplinary Research, Alexandru Ioan Cuza University of Iasi, 11 Carol I Bd., 700506 Iasi, Romania; violeta.mangalagiu@uaic.ro; 4Faculty of Food Engineering, Stefan cel Mare University of Suceava, 13 Universitatii Str., 720229 Suceava, Romania

**Keywords:** Alzheimer’s disease, amyloid-beta peptides, β-sheet breaker potential, neuropeptide-based drugs, pharmacological activity, solid-phase peptide synthesis, matrix-assisted laser desorption/ionization time-of-flight mass spectroscopy, circular dichroism

## Abstract

Our work discusses the investigation of 75 peptide-based drugs with the potential ability to break the β-sheet structures of amyloid-beta peptides from senile plaques. Hence, this study offers a unique insight into the design of neuropeptide-based drugs with β-sheet breaker potential in the amyloid-beta cascade for Alzheimer’s disease (AD). We started with five peptides (^15^QKLVFF^20^, ^16^KLVFF^20^, ^17^LVFF^20^, ^16^KLVF^19^ and ^15^QKLV^18^), to which 14 different organic acids were attached at the N-terminal. It was necessary to evaluate the physiochemical features of these sequences due to the biological correlation with our proposal. Hence, the preliminary analysis of different pharmacological features provided the necessary data to select the peptides with the best biocompatibility for administration purposes. Our approaches demonstrated that the peptides ^17^LVFF^20^, NA-^17^LVFF^20^, ^16^KLVF^19^ and NA-^16^KLVF^19^ (NA-nicotinic acid) have the ability to interfere with fibril formation and hence improve the neuro and cognitive functions. Moreover, the peptide conjugate NA-^16^KLVF^19^ possesses attractive pharmacological properties, demonstrated by in silico and in vitro studies. Tandem mass spectrometry showed no fragmentation for the spectra of ^16^KLVF^19^. Such important results suggest that under the action of protease, the peptide cleavage does not occur at all. Additionally, circular dichroism confirmed docking simulations and showed that NA-^16^KLVF^19^ may improve the β-sheet breaker mechanism, and thus the entanglement process of amyloid-beta peptides can be more effective.

## 1. Introduction

Alzheimer’s disease (AD) is a sporadic and hereditary neurodegenerative illness, which causes amnestic cognitive impairment in its archetypal form and non-amnestic cognitive impairment in its less prevalent variations [1]. Furthermore, neural atrophy and disrupted inter-neuronal connections can occur [2]. By 2050, it is thought that up to 107 million people will be affected throughout the world [3]. Furthermore, AD has become incurable and life-threatening and imposes a significant socioeconomic cost due to its multifactorial nature [4]. The biological and environmental underpinnings of AD, particularly its correlation with amyloid- and Tau-protein-related systems, have been revealed recently through the research of this topic. This paradigm is applicable to autosomal dominant AD but not to sporadic AD [5]. However, the precise molecular events and biological pathways underlying the disease have yet to be discovered [6].

One of the most difficult tasks in AD research is to assemble the vast framework of scientific evidence regarding the disorder, much of which appears to be contradictory, into a coherent and credible pathogenic process [7]. Within this context, therapeutic efforts are still trying to uncover targets that can significantly alter the clinical course of people with Alzheimer’s disease [1]. Among existing theories that characterize the development and evolution of AD, the amyloid hypothesis has been a long-term adherent to AD due to the involvement of various types of amyloid-beta peptides (Aβ) in the impairment of neuronal and cognitive function [4]. Emerge of extracellular-amyloid accumulation in the form of neuritic plaques and intracellular accumulation of hyperphosphorylated tau in the type of neurofibrillary tangle continues to be the main neuropathologic criterion for the diagnosis of AD [8]. Moreover, a recent study showed that the extensive collaborative linkages between Aβ and innate immunity’s established anti/pro-inflammatory cytokines allow amyloid misfolding to be included in the growing immunopathic disease mechanism theories of AD [9]. In this scenario, Alzheimer’s symptoms develop, and the individuals show significant cognitive impairments [10].

Many researchers have refocused on finding a distinct molecular assembly of defined size, which describes the key cause of AD after the amyloid cascade hypothesis was reformulated. The main focus of this hypothesis was related to oligomeric aggregates of Aβ as the primary toxic species causing AD [11,12]. It was also showed that the Aβ oligomers have noxious effects on post-synaptic protein Neuroligin-1 (NLGN1), which has critical roles in synapse formation, maturation, plasticity and maintenance [13]. The most common forms of Aβ in the human brain were proved to be Aβ_(1–40)_ and Aβ_(142)_. However, the Aβ_(25–35)_ fragment, which is physiologically predominant in aged individuals, has just been revealed to play a key role in AD due to its particular aggregation characteristics [14,15].

Furthermore, it was noticed that there is a clear connection between the development of amyloid fibrils and the decrease in the native state’s conformational stability of these peptides. The Aβ peptides in AD are inherently unfolded. Hence, a fibrillar aggregate with a rich β-sheet structure was identified in Alzheimer’s disease deposits [16,17]. Despite a large number of studies, the underlying mechanisms that cause the Aβ aggregates to break the β-sheet structure remain unknown [18]. Therefore, the eradication of the Aβ peptide aggregates that form amyloid fibrils remains a complex task in the development of an anti-AD therapeutic strategy.

Drug discovery efforts in Alzheimer’s disease face a unique set of hurdles [19]. Currently, altered biochemistry of the Aβ cycle is thought to be a major biological characteristic of AD and a possible therapy target [20]. In addition, peptide-based drugs can represent a viable solution to alleviate the symptoms of various diseases. Due to their low intrinsic toxicity, peptides are particularly appealing. The use of highly effective and precise peptide drugs would considerably help neurological conditions such as neurodegeneration, pain, psychiatric disorders, stroke and brain tumors [21,22,23]. As a result of advancements in biotechnology, a wide range of peptide drugs are currently produced on a commercial basis. However, because of limited absorption from the gastrointestinal tract, the majority of these therapeutic peptides are still delivered by parenteral method [24]. In addition to the parenteral route, the peptide-based drugs can also intranasally be administered [25]. Thus, with a biocompatibility closer to gastrointestinal or blood–brain barrier (BBB) administration, these treatments can be more effective in evaluating their intranasal or cutaneous pharmacological properties.

Nevertheless, there has been an increase in research of peptide-based drug applicability in various diseases. More than 40 cyclic peptide drugs derived from natural compounds have been authorized in clinical trials during the last few decades [26]. Recently, the molecular way in which neuroprotective NAP peptide (^1^NAPVSIPQ^8^) and its analog with acetylsalicylic acid can stabilize axonal microtubules has been proved [27,28,29]. It was also reported that the semi-elastic lipid nanoparticles significantly enhanced peptide oral availability and could be a viable future oral peptide delivery technology for peptide medicines [30]. Furthermore, several toxin-derived peptides have been developed into pharmaceuticals which are used to treat diabetes, hypertension, chronic pain and other disorders [31]. Moreover, a similar study presents a critical theoretical foundation as well as significant empirical data on the essential physicochemical aspects of anticancer peptide-based drug development [32].

Different in vitro studies performed by Tjerbenberg and his team suggested that during Aβ polymerization and fibril formation, Aβ_(16–20)_ residues act as a binding sequence. This pentapeptide, ^16^KVLFF^20^, may be used as ligands that bind to Aβ and prevent the formation of amyloid fibrils [33,34]. Thus, it may break the β-sheet structure of amyloid fibrils by intercalation among the Aβ monomers. It was also observed that the sequence 15–20 of Aβ represents the crucial fragment which forms β-sheet structures [35]. Therefore, an important step to prevent the development of Aβ fibrils could be to break the interaction between two fragments of Aβ_(15–20)_. On the other hand, a common problem in peptide-based drugs relates to the BBB and gastrointestinal administration specificity. Hence, these drugs must meet the criteria regarding the lipophilicity and molecular weight according to pharmacological rules [36].

Therefore, in the present study we investigated over 70 new peptide-based drugs (peptide conjugates based on the Aβ_(15–20)_ sequence) in respect to the lipophilicity and biocompatibility of different methods of administration (e.g., intranasal). We utilized a particular set of peptide conjugates never studied before.

## 2. Results

### 2.1. Analysis of Pharmacological Properties

Many drug development shortcomings can be attributed to inadequate pharmacokinetics and bioavailability, in addition to their efficacy and toxicity. Two pharmacokinetic features that must be estimated at various stages of the drug development process are gastrointestinal absorption and brain access. Therefore, a method called the brain or intestinal estimated permeation predictive model (BOILED-Egg) was recently proposed in order to investigate pharmaco-properties of various proposed drugs. This method was tested with an accuracy of over 90% in the case of commercial drugs [37]. Furthermore, BOILED-Egg is based on topological surface area or polarity (tPSA/Å) and lipophilicity (WLOGP) [38,39]. For efficient gastrointestinal absorption of a drug, the commonly accepted limits for tPSA/Å are lower than 142 Å, and the value of WLOGP should be between −2.3 and 6.8. In the case of the BBB, the polarity parameter must be lower than 79 Å and the relative lipophilicity from 0.4 to 6.0 [37]. Therefore, drugs that possess these values have a high probability to access the central nervous system through the gastrointestinal or intranasal administration.

In our investigation of pharmacological administration properties, we applied this approach for each peptide-based drug designed in Table 1. The polarity and lipophilicity were considered with the purpose of obtaining a histogram of effective administration for the human body regardless of intranasal, gastrointestinal or BBB biocompatibility. The polarity based on the topological surface for each conjugate is presented in Figure 1a and lipophilicity in Figure 1b.

According to Figure 1, there are significant differences between the pharmacological features of these peptide-based drugs. Regarding polarity, the histogram presented in Figure 1a shows that the best biocompatibility is possessed by peptides 3 (^17^LVFF^20^) and 4 (^16^KLVF^19^) from the green square, while the peptides 1 (^15^QKLVFF^20^), 2 (^16^KLVFF^20^) and 5 (^15^QKLV^18^) present the highest tPSA/Å value. These noteworthy differences may be due to the structural properties. ^15^QKLVFF^20^ and ^16^KLVFF^20^ with five amino acid residues display a larger molecular surface, while the tetrapeptide with the glutamine residue (peptide 5) possesses an extra amino group compared with ^17^LVFF^20^ and ^16^KLVF^19^. Moreover, the presence of additional oxygen or nitrogen atoms seems to increase the polarity of the peptides and therefore to develop incompatibility with administration purposes. Furthermore, the lipophilicity parameter presented in Figure 1b indicates that the peptide conjugates with palmitoyl (3) and stearoyl acid (6) show values above the limit, while the others exhibited peptide–drug display values within the set values. This could suggest that conjugates with a higher number of carbon atoms (more than 16) are not suitable for peptide coupling according to this hypothesis.

Hence, following this preliminary analysis, the peptides with the best biocompatibility regarding intranasal, gastrointestinal and BBB administration were based on the sequence ^17^LVFF^20^ and ^16^KLVF^19^. In addition, the conjugates with palmitoyl and stearoyl acid were not recommended to be synthesized due to the poor cross-barrier administration.

Secondly, these 75 peptides were subjected to another theoretical approach based on six physicochemical marks: lipophilicity, size, polarity, insolubility, instauration and flexibility. The limits of these parameters for optimal administration were set according to Section 2.1. Our purpose for this analysis was to investigate these physicochemical properties in order to obtain information regarding the biostructural compatibility for drug design in the case of oral administration. The representations of physicochemical marks regarding lipophilicity, size, polarity, insolubility, instauration and flexibility of the peptides with the corresponding organic compound described in Table 1 are presented in Figure 2.

Figure 2 shows the interesting distribution of the physicochemical properties for drug delivery in the case of each conjugate. The parameters which are constant out of the assigned limit for oral administration are flexibility, size and polarity. However, due to the structural characteristics of peptides, these three parameters are expected to exceed the values described in Section 2. The peptides’ flexibility and size could not be constrained by chemical compound coupling, while the polarity represents an additional feature which could be improved for drug delivery during SPPS.

Hence, according to Figure 2a, these five peptides present a physicochemical distribution in the accepted limits in the case of lipophilicity, instauration and solubility (green hexagon), while the coupled peptides show variable values. The conjugates with lauric acid (1), palmitoyl acid (3), decanoic acid (5), stearoyl acid (6), 4-hydroxy-3-methoxymandelic acid (8), naphthaleneacetic acid (11), salicylic acid (13) and acetylsalicylic acid (14) exhibit a wide distribution, which was not compatible with drug delivery, while hexanoic acid (2), octanoic acid (4), taurine (7), indole-3-butyric acid (9), 3,5-dinitrosalicylic acid (10) and nicotinic acid (12) display an acceptable dispersion, compatible with oral administration.

### 2.2. Bioactivity Screening

According to the results outlined in Section 2.1, peptides with the best acceptable biocompatibility with the usual drug administration were peptide 3 (^17^LVFF^20^) and peptide 4 (^16^KLVF^19^). In addition, regarding the chemical compounds used for conjugation, between hexanoic acid, octanoic acid, taurine, indole-3-butyric acid, 3,5-dinitrosalicylic acid and nicotinic acid, which presents an acceptable dispersion compatible with oral administration, we chose to further investigate the nicotinic acid-peptide (3-12 and 4-12). The fatty acids taken into account, hexanoic acid (2) and octanoic acid (4), have no additional therapeutic effects, while for indole-3-butyric acid (9), even this plant growth regulator poses no known risks to humans but has no neuroprotective outcome [40]. Nicotinic acid (NA), also known as niacin, regulates the cholesterol level and suppresses the mitogen-activated protein kinase pathway and attenuates brain injury after cardiac arrest in rats [41]. Hence, we further chose to investigate our designed drugs based on the sequence of the peptide 3 (^17^LVFF^20^) and peptide 4 (^16^KLVF^19^) with nicotinic acid.

Using the bioactivity screening (see the methodology described in Section 2), we investigated which class of active biomolecules from the human body, such as proteins, enzymes and receptors, could interact with these proposed drugs. Our purpose was to analyze the possible influence on the therapeutic effect of our peptides. These interactions with peptides and peptide conjugates 3 (^17^LVFF^20^), 3-12 (NA-^17^LVFF^20^), 4 (^16^KLVF^19^) and 4-12 (NA-^16^KLVF^19^), listed by each class of biomolecules based on probability, are presented in Figure 3.

According to Figure 3, there are several classes of biomolecules which could interact with ^17^LVFF^20^ and ^16^KLVF^19^, including their conjugates with NA. Figure 3a shows that the highest probability of interactions for ^17^LVFF^20^ is with protease (28%), followed by Family AG protein-coupled receptor (24%). After coupling, the probability for this peptide conjugate (NA-^17^LVFF^20^) drastically increases for Family AG protein-coupled receptor (52%), while the probability for protease was the same.

Additionally, in the case of ^16^KLVF^19^ (Figure 3b), the value obtained for protease was 32%, whereas for NA-^16^KLVF^19^, it was raised at 32%. Regarding Family AG protein-coupled receptor, the probability remained similar following coupling (from 32% to 36%).

Therefore, these results suggest that the addition of NA increases the affinity for these classes of biomolecules. Furthermore, an improvement of probability interaction with Family AG protein-coupled receptor, as observed in the data above, could represent an important step for neuroprotective functions. It was proved that the overexpression of Family AG protein-coupled receptor inhibits oxidative toxicity in the brain [42]. Moreover, humanin, a newly identified neuroprotective factor, uses the G protein-coupled formylpeptide receptor-like-1 as a functional receptor [43]. In addition, by increasing the activity of proteases, they may catalyze the cleavage for amyloid-beta peptides (1–42) and (1–40), monomer and fibrils.

In addition, we performed pharmacokinetic investigations of cytochrome P450 (CYP) in the presence of our designed peptides in order to improve the knowledge regarding biocompatibility and various interactions. Thus, predicting the probability that a molecule will cause inhibition of CYPs represents a critical step for drug development. Furthermore, such an isoenzyme superfamily possesses a crucial role in drug clearance via metabolic biotransformation. Hence, the inhibition of these isoenzymes could provide us supplementary information on the major cause of pharmacokinetics-related drug–drug interactions. Therefore, due to the decreased clearance and buildup of the peptide-based drugs or their metabolites, this inhibition could cause toxic or other undesired side effects. For example, it has been proposed that CYP can metabolize small molecules in a synergistic manner in order to promote tissue and organism protection [44]. Thus, using the same SwissADME computational method, we investigated the estimation for our designed peptides (presented in Figure 3) to be inhibitors of the most important CYP isoenzymes: CYP1A2, CYP2C19, CYP2C9, CYP2D6 and CYP3A4.

Hence, the support vector machine algorithm (SVM) was applied, and if the models returned yes/no, the analyzed molecule possessed a higher probability to be an inhibitor of a selected isoenzyme [45,46]. The specific methodology applied in our case was in the best agreement with the aforementioned literature [45]. Consequently, our results for the synthetized peptides except NA-^17^LVFF^20^ showed negative response, and therefore these molecules could not inhibit the CYP superfamily. Nevertheless, only NA-^17^LVFF^20^ could inhibit CYP3A4, but a study performed by Shou et al. showed that there is evidence for the simultaneous binding of two substrates in a cytochrome P450 active site, and 7,8-benzoflavone could reactivate CYP3A4 substrate [47].

### 2.3. Structural Interaction and Docking

In order to investigate the potential interaction of NA-^17^LVFF^20^ and NA-^16^KLVF^19^ with amyloid-beta monomer and fibrils, we utilized Flexible Alignment (Figure 4). Starting from our hypothesis, the purpose of this approach was to study the structural interaction of these molecules in order to break the formation of β-sheet conformations. In addition, by promoting the binding between NA-^17^LVFF^20^ and NA-^16^KLVF^19^ with amyloid-beta monomer, the probability to form fibrils and consequently β-sheet structures could decrease drastically (Figure 4a).

Figure 4c,d shows different alignment patterns of our designed peptides and Aβ_(1–42)_. Compared with Figure 1b, there are significant differences regarding amyloid-beta monomer conformation. These results suggest that our short peptide entangles to the primary sequence of Aβ_(1–42)_, simultaneous with α-helix motif inhibition. In this regard, the amyloid-beta changes its conformations to a random coil structure, which could increase the entanglement process. Moreover, following addition of NA to the ^17^LVFF^20^ and ^16^KLVF^19^, a progression of α-helix motif inhibition was noticed. Therefore, this may suggest that the peptide conjugates NA-^17^LVFF^20^ and NA-^16^KLVF^19^ could develop the entanglement process and hence may improve the process of the β-sheet breaker mechanism.

Furthermore, in order to investigate the potential effect of these peptides for β-sheet breaker on the amyloid-beta fibrils, we utilized docking simulations. The docking results, according to the methodology described in Section 2.3, are presented in Figure 5. In addition, van der Waals interactions between ^17^LVFF^20^, NA-^17^LVFF^20^, ^16^KLVF^19^ and NA-^16^KLVF^19^ with Aβ_(1–42)_ fibrils are presented in Figure A1 and the structural interactions of the same systems in Figure A2.

According to Figure 5, the molecular pathway through which peptides interact with Aβ_(1–42)_ fibrils seems to be at the top (Figure 5a–c) or at the bottom (Figure 5d) of the fibril. Interestingly, this suggests that if the interactions mainly occur at the end of the amyloid-beta fibril structure, there is no possibility of binding another Aβ_(1–42)_ monomer in order to increase the length of the fibril due to this particular blocking. Hence, the process of the β-sheet breaker could occur. Moreover, if in a therapeutic pathway a combination between ^16^KLVF^19^ and NA-^16^KLVF^19^ is possible, where the first peptide could be attached at the top (Figure 5c), and the peptide-based drug at the end of the fibril (Figure 5d) suggests that fibrillar formation obstruction could ensue.

Furthermore, van der Waals interactions between ^16^KLVF^19^ and Aβ_(1–42)_ fibrils show that the peptide possesses the ability to reach the terminal core of the fibril structure by generating a pocket by reaching the oxygen-rich (carboxyl groups) and nitrogen-rich structures (amino groups), and therefore it could further destabilize it (Figure A1c). In addition, the other peptides seem to interact longitudinally with amyloid-beta (Figure A1a,b,d). The structural binding shows that tyrosine could act as a stabilizer through its phenyl residue by generating π-H arene–cation interactions (Figure A2).

### 2.4. Solid-Phase Peptide Synthesis (SPPS)

In order to investigate experimentally the structures of these peptides, we utilized solid-phase peptide synthesis. In addition, the chemicals with analytical grade were obtained from commercial sources, and the solutions were prepared using ultrapure water from a Millipore purification system. The peptides ^17^LVFF^20^ and ^16^KLVF^19^ and the peptide-based drugs coupled with nicotinic acid, NA-^17^LVFF^20^ and NA-^16^KLVF^19^ were synthesized according with the experimental protocol described in Section 2.4. The crude peptide following synthesis was further investigated and separated by RP-HPLC. Consequently, the purified fractions that resulted from chromatography were subjected to MALDI-ToF and tandem mass spectrometry.

### 2.5. Reverse-Phase High-Performance Liquid Chromatography (RP-HPLC)

The crude peptide obtained following SPPS was subjected to Reverse-Phase High-Performance Liquid Chromatography in order to obtain the fractions with the highest purity (Figure A3). Furthermore, these fractions were injected again in Dionex UltiMate 3000 UHPLC Focused to confirm the peptide purity (Figure 6). The methodology used to separate these peptides is described in Section 2.

According to Figure 6, a difference in the hydrophobicity feature of the peptides was noticed. This could be due to the structural characteristic of each conjugate (Figure A4). Hence, ^16^KLVF^19^ demonstrated the shortest retention time because of the lysine, a more hydrophilic residue than phenylalanine from ^17^LVFF^20^ (Figure A4a,c). In addition, following the NA coupling, the retention time increased for both ^16^KLVF^19^ and ^17^LVFF^20^, due to the hydrophobicity of this acid (Figure A4b,d).

Although, the initial purity of the newly synthesized peptides was estimated to be lower than 85% based on peak area of the chromatograms calculated from Figure A3 (Table A2), only RP-HPLC fractions with a purity of 99% were used in subsequent investigations (Figure 6).

### 2.6. MALDI-ToF/ToFMass Spectrometry

Following the RP-HPLC separations, the fractions with the highest purity were subjected to matrix-assisted laser desorption/ionization time-of-flight mass spectrometry (MALDI-ToF-MS) analysis, performed with a Shimadzu Axima Performance MALDI-ToF spectrometer. The samples were co-crystallized with α-CHCA and further analyzed in the positive reflectron mode. The MS spectra of the synthesized peptides are shown in Figure 7.

According to Figure 7, besides a molecular ion for each peptide ([M + H]^+^), sodium and potassium adducts were also noticed. Hence, Figure 7a shows two additional signals, at 546.8 Da for the ^17^LVFF^20^ adduct with Na^+^ and at 562.8 Da for that with the K^+^ ion. Regarding the conjugate with NA, the same adducts were observed at 651.6 Da for NA-^17^LVFF^20^ with sodium and at 667.6 Da for the potassium adduct (Figure 7b). In the case of peptide 4 (^16^KLVF^19^), similar signals were found in the spectra (Figure 7c,d), being specific both for Na^+^ (at 527.6 and 632.6 Da) and K^+^ adducts (543.6 and 648.6 Da).

In order to investigate the structural stability of these peptides, we utilized tandem mass spectrometry through collision-induced dissociation in a positive reflectron mode. The main purpose of this approach was to analyze the strength of the bonds between amino acids in our designed structures. Due to biomolecule activity in the human body, some proteases may cleave these peptides. Therefore, the MS/MS data could show the location of the weaker bonds and hence indicate where the cleavage may occur. The stronger the bonds, the more stable the peptide, and, as a result, it can withstand the cleavage process. Thus, the peptides could display their therapeutic effect completely. The mass spectra of the ^17^LVFF^20^ and ^16^KLVF^19^ peptides following CID, are presented in Figure 8.

Tandem mass spectrometry of the newly designed peptides shows fascinating data. According to Figure 8a, five signals for ^17^LVFF^20^, including the molecular ion at 524.6 Da ([M + H]^+^), were observed. The fragmentation occurred in this particular case with the formation of y^1+^ at 312.6 Da, b^2^ at 360.7 Da, a^1^ at 479.8 Da and b^3^ at 507.8 Da. This suggests that following the CID, the weaker bonds are found between phenylalanine residues, resulting in generation of the ^17^LV^18^ fragment (the inset representation from Figure 8a).

However, in the case of the ^16^KLVF^19^ peptide, the spectrum did not exhibit any fragmentation, even when the highest power pulsed per shot within the MALDI-ToF device was used (Figure 8b). These interesting data suggest that the peptide possesses very strong bonds, which improve biostability. Hence, with the protease activity, peptide cleavage may not occur at all. The drug based on this sequence conjugate with NA could significantly improve the neuroprotective action. Therefore, in this particular case, the peptide ^16^KLVF^19^ could attach at the Aβ_(1–42)_ monomer and fibrils, thus inhibiting fibril formation. Moreover, it could improve the β-sheet breaker mechanism, while with the release of NA (D), other neuroprotective functions could be improved (the inset representation from Figure 8b).

### 2.7. Theoretical Circular Dichroism

In order to investigate the peptide conformations and to study the effect of these motifs on the β-sheet breaker mechanism, we utilized a theoretical circular dichroism (CD). The CD spectra of ^17^LVFF^20^, NA-^17^LVFF^20^, ^16^KLVF^19^ and NA-^16^KLVF^19^ peptides are displayed in Figure 9.

Figure 9 shows interesting wavelengths regarding peptide conformations. According to these spectra and the data obtained with reference to the α-helix, β-sheet and random coil motif distribution (Table A3), the addition of NA modifies the peptide conformations. In the case of ^17^LVFF^20^ and NA-^17^LVFF^20^, only a few differences were observed. Nevertheless, when it comes to the ^16^KLVF^19^ peptide, the presence of NA drastically increases the random coil motif, from 32.34% to 56.67%. This suggests that the NA-^16^KLVF^19^ confirms docking simulations and therefore could improve the β-sheet breaker mechanism. With a higher percentage of random coil structure of NA-^16^KLVF^19^, the interaction with Aβ_(1–42)_ monomer might be stronger, and hence the entanglement process can be more effective than the ^16^KLVF^19^ peptide without NA.

## 3. Discussion

In this study, we investigated the potential activity of 75 peptide-based drugs for the β-sheet breaker mechanism. In order to increase the biocompatibility for different methods of administration, we started with five peptides based on the Aβ_(15–20)_ sequence to which 14 different organic compounds were attached at the N-terminal of the sequence.

Following the preliminary analysis of different pharmacological features, the peptides with the best biocompatibility regarding intranasal, gastrointestinal and BBB administration were based on the sequence ^17^LVFF^20^ and ^16^KLVF^19^. In addition, we chose to investigate the nicotinic acid-peptide conjugate because it can regulate different forms of neuroprotective actions. Furthermore, it was noticed that that the addition of NA to the peptide molecule increases the affinity for Family AG protein-coupled receptor and for proteases, which could also improve the cognitive functions in Alzheimer’s disease.

Moreover, according to Flexible Alignment data, the peptide conjugates NA-^17^LVFF^20^ and NA-^16^KLVF^19^ developed the entanglement process, and hence they could improve the process of the β-sheet breaker mechanism. In addition, the molecular pathway through which peptides interact with Aβ_(1–42)_ fibrils seems to be at the top or at the bottom of the fibril. Hence, the process of the β-sheet breaker could occur. Moreover, if a combination between ^16^KLVF^19^ and NA-^16^KLVF^19^ is used in a therapeutic pathway, where the first peptide is attached at the top and the peptide-based drug is at the end of the fibril, fibrillar formation obstruction could ensue. The van der Waals interactions between ^16^KLVF^19^ and Aβ_(1–42)_ fibrils show that the peptide might possess the ability to destabilize the structures.

Following preliminary investigation, four conjugates from the investigated line-up were synthetized using solid-phase peptide synthesis. Tandem mass spectrometry of the newly designed peptides showed, following the CID, weaker bonds between phenylalanine residues, which result in the formation of the ^17^LV^18^ fragment. In the case of the ^16^KLVF^19^ peptide, the spectrum did not evidence any fragmentation. These interesting data suggest that, in the presence of a protease, the peptide cleavage is not possible. This could significantly improve the neuroprotective action. Therefore, the peptide ^16^KLVF^19^ could attach at the Aβ_(1–42)_ monomer and fibrils, inhibiting fibril formation and improving the β-sheet breaker mechanism, with the release of NA to improve neuroprotective functions.

Consequently, CD spectra show that ^16^KLVF^19^ linked to NA drastically increases the proportion of random coil conformers. This suggests that the NA-^16^KLVF^19^ confirms docking simulations and, therefore, could improve the β-sheet breaker mechanism. Moreover, the entanglement process can also be effective.

The novelty of our work is represented by the study of new combinations between Aβ_(15–20)_ and various organic acids, as well as a unique computational estimate of these properties. However, it is necessary to further investigate the in vitro and in vivo properties of peptide-based drugs. In addition, this work focused on the main features of these beta-sheet breaker molecules when the main design was approached. Moreover, a recent study showed that the ^1^FRSAPFIE^8^ (FRS) octapeptide could act as a drug delivery system [48]. Hence, a system formed by our beta-sheet breaker peptides and FRS might act as a transporter (e.g., intranasally) for our final purpose: the disruption of amyloid beta fibril formation.

## 4. Materials and Methods

We started with five peptides (^15^QKLVFF^20^, ^16^KLVFF^20^, ^17^LVFF^20^, ^16^KLVF^19^ and ^15^QKLV^18^), to which 14 different organic compounds were attached at the N-terminal sequence of the peptides (lauric acid, hexanoic acid, palmitoyl acid, octanoic acid, decanoic acid, stearoyl acid, taurine, 4-hydroxy-3-methoxymandelic acid, indole-3-butyric acid, 3,5-dinitrosalicylic acid, naphthaleneacetic acid, nicotinic acid, salicylic acid and acetylsalicylic acid). Due to the structural nature of fatty acids, each selected organic compound from this classification (lauric acid, hexanoic acid, palmitoyl acid, octanoic acid, decanoic acid, stearoyl acid) could increase the lipophilicity of the synthesized peptide-based drugs. Hence, the fatty acid might assist a better BBB permeability, as reported by Hamilton et al. [49]. In addition, taurine directly binds to Aβ oligomers and recovers cognitive deficits in Alzheimer’s model mice [50]. Furthermore, the benzene ring from 4-hydroxy-3-methoxymandelic acid, indole-3-butyric acid, 3,5-dinitrosalicylic acid and naphthaleneacetic acid could also develop the lipophilicity feature. Moreover, nicotinic acid, salicylic acid and acetylsalicylic acid have been previously reported as possessing neurobehavioral recovery and brain plasticity proprieties [51,52,53].

Following preliminary analysis, we synthetized four conjugates from the investigated line-up using solid-phase peptide synthesis (SPPS), based on the peptides with the best administration biocompatibility. Consequently, these peptides were purified using reverse-phase high-performance liquid chromatography (RP-HPLC) and characterized by matrix-assisted laser desorption/ionization time-of-flight mass spectrometry (MALDI-ToF MS), tandem MS and Circular Dichroism (CD).

### 4.1. Estimation of the Pharmacological Properties

In order to investigate the biocompatibility of 75 potential β-sheet breaker peptide-based drugs, we utilized various computational methods. First, five peptides (^15^QKLVFF^20^, ^16^KLVFF^20^, ^17^LVFF^20^, ^16^KLVF^19^ and ^15^QKLV^18^) were designed based on the 15–20 sequence of Aβ, to which 14 different organic compounds were attached at the N-terminal of the peptide (lauric acid, hexanoic acid, palmitoyl acid, octanoic acid, decanoic acid, stearoyl acid, taurine, 4-hydroxy-3-methoxymandelic acid, indole-3-butyric acid, 3,5-dinitrosalicylic acid, naphthaleneacetic acid, nicotinic acid, salicylic acid and acetylsalicylic acid). These neuropeptide-based drugs coupled with each organic compound are systematically presented in Table 1.

Here, the brain or intestinal estimated permeation predictive model (BOILED-Egg) was used in order to investigate the pharmaco-properties of our designed peptides and conjugates [37]. This approach was based on topological surface area (polarity—tPSA/Å) and lipophilicity (WLOGP) of these molecules [38,39]. For efficient gastrointestinal absorption of a drug, the commonly accepted limits for tPSA/Å are lower than 142 Å, and the value of WLOGP should be between −2.3 and +6.8. In the case of the BBB, the polarity parameter must be lower than 79 Å with a relative lipophilicity from 0.4 to 6.0 [37]. In order to fit these pharmacological parameters, the sequence of the designed peptides was introduced into a computational system developed by Swiss Institute of Bioinformatics [45].

Furthermore, to analyze efficacy of the conjugates, these peptides were subjected to the same computational method, following six physicochemical features: lipophilicity, size, polarity, insolubility, instauration and flexibility. The limits of these parameters for optimal oral administration were set as it follows: In the case of the lipophilicity descriptor according to the octanol–water partition coefficients, the value was set between −0.7 and 5.0, molecular weight between 150 and 500 g/mol, tPS/Å—from 20 to 130 Å, solubility based on molar solubility in water (with ESOL model, log S) lower than 6, instauration (which describes the fraction of carbon in sp^3^ hybridization) between 0.25 and 1 and in the case of flexibility, the molecule should have no more than nine rotatable bonds [54,55,56].

### 4.2. Bioactivity Screening

Consequently, the selected peptides were subjected to bioactivity screening in order to investigate the capability to bind to different macromolecular targets from *Homo sapiens*. This method was essential for deciphering the molecular pathways underlying certain phenotypic or biocompatibility, rationalizing potential side effects, predicting off-target repercussions and evaluating the feasibility of therapeutically relevant molecules [57]. The SwissTargetPrediction algorithm used in this particular case was based on the similarity principle. This principle asserts that two similar molecules are likely to have similar characteristics [58].

### 4.3. Structural Interaction and Docking

Subsequently, the interaction between our designed peptides and the amyloid-beta monomer (1–42) obtained from the Protein Data Bank (1IYT) was analyzed [59]. This crystallographic structure was built and subjected to Flexible Alignment in Molecular in Operating Environment 2016.02 software (MOE) developed by Chemical Computing Group [60]. In addition, force field charges were calculated prior to the search of these interactions. The similarity terms were set according to the H-Bond Donor Projection, H-Bond Acceptor Projection and to the Aromatic Centers. Furthermore, the iteration limit was set to 1000 and the failure limit to 50. The energy cutoff used for this approach was set to 15 [61,62,63].

Moreover, the ligand interaction was used to depict the active sites of amyloid-beta peptides with our designed molecules. These interactions were obtained with MOE software using the methodology available in the literature [64]. A series of steps was followed regarding this protocol: The ligand was set in the center of the diagram, followed by determination of the ligand’s coordinates, the residues that form strong H-bonds with the ligand were set, the residues close to the ligand with weaker interactions were placed and the solvent accessibility and proximity contour were calculated [64,65,66].

SwissDock, a computational tool, was used to dock the selected peptides to the binding sites of near-atomic resolution fibril structure of complete amyloid-beta (142) obtained by cryo-EM (Protein Data Bank: 5OQV) [67]. SwissDock was developed using the EADock DSS docking software, which applies the CHARMM energy estimate algorithm to generate binding modes. Subsequently, the binding potential with the most favorable energies was evaluated with FACTS and clustered the system [68,69]. Furthermore, Chimera 1.14 and MOE software were used to analyze the docking pattern [70].

### 4.4. Solid-Phase Peptide Synthesis (SPPS)

The chemicals with analytical grade were obtained from commercial sources, and the solutions were prepared using ultrapure water from a Millipore purification system (Milford, MA, USA). Fmoc-Lys(Boc)-OH, Fmoc-Leu-OH, Fmoc-Val-OH and Fmoc-Phe-OH and piperidine were purchased from Fluka (UK). The Fmoc-Rink-Amide-(aminomethyl)-Resin (0.48 mmol/g, 200–400 mesh, 1% DVB) was obtained from Intavis Bioanalytical Instruments (Koln, Germany) and benzotriazole-1-yl-oxy-tris-pyrrolidino-phosphonium hexafluorophosphate (PyBOP) from Novabiochem (Rosh Ha’ayin, Israel). *N*,*N*-dimethylformamide (DMF), diethyl ether, dichloromethane, acetonitrile (AcCN) and ethanol were purchased from Roth (Karlsruhe, Germany), whereas acetic acid, trifluoroacetic acid (TFA) and *N*-methyl-morpholine (NMM) were from Merck (Darmstadt, Germany). Triisopropylsilane (TIS) and α-cyano-4-hydroxycinnamic acid (α-CHCA) were obtained from Sigma Aldrich (Gillingham, UK), while the nicotinic acid was purchased from Chinoin (Budapest, Hungary). In addition, the bromophenol blue indicator was bought from Scharlau (Hamburg, Germany).

For each synthesis, we determined the quantity of protected amino acids and reagents required to obtain 250 µM of peptide-based drugs. Thus, the resin was placed in a 2 mL syringe reactor and swelled for 30 min with DMF, followed by deprotection with a piperidine/DMF solution (1:4). The washing steps were performed using DMF. The amino acid coupling was achieved with NMM and PyBOP for 50 min at room temperature (25 °C). The coupling step was repeated twice for each amino acid. Following peptide synthesis, nicotinic acid (NA) was added similarly with Fmoc-amino acid coupling, except for the incubation time, which was 2 h instead of 50 min. Furthermore, bromophenol blue test was used to monitor the coupling reaction [71]. The synthesis was mixed by stirring, and the resin was filtered with polyethylene.

The peptide cleavage from the resin was performed with a solution of TFA, TIS and ultrapure water (95:2.5:2.5, *v*/*v*/*v*). Consequently, the peptides were precipitated with 5% acetic acid (in ultrapure water) and subjected to lyophilization (Liophilizer Alpha 1-2 LD Plus 101521, Martin Christ, Osterode am Harz, Germany).

### 4.5. Reverse-Phase High-Performance Liquid Chromatography (RP-HPLC)

The Reverse-Phase High-Performance Liquid Chromatography (RP-HPLC) was performed using a Dionex UltiMate 3000 UHPLC Focused from Thermo Scientific (Bremen, Germany). In order to achieve chromatographic separation of peptides, a 250 × 4.6 mm analytical C18 polymeric bonding column was used (Vydac 218MS, 5 µm, 300 Å, SCP, Grace Davidson Discovery Science, Bannockburn, IL, USA) along with Chromeleon 7.2.6 software for data analysis. The wavelength was set to 220 nm in order to monitor the peptide bond [27].

The gradient (solvent A: 0.1% TFA and solvent B: 80% AcCN in 0.1% TFA, *v*/*v*) used to separate the synthesized peptides (flow: 1 mL/min, injection volume: 10 µL from a 1 mg/mL peptide in 0.1% TFA: 80% AcCN in 0.1% TFA—95:5, *v*/*v*, solution) is shown in Table A1.

### 4.6. MALDI-ToF Mass Spectrometry

Matrix-assisted laser desorption/ionization time-of-flight mass spectrometry (MALDI-ToF-MS) analysis was performed using a Shimadzu Axima Performance MALDI-ToF/ToF spectrometer from Shimadzu Group Company. In order to measure MS and collision-induced dissociation (CID—tandem mass spectrometry) spectra of the peptides, the instrument was operated in the positive reflectron mode. The samples were co-crystallized with an organic matrix (a saturated suspension of α-CHCA in a solvent mixture 2:1—AcCN:0.1% TFA in water type 1) in a ratio of 1:1 and further prepared with the dry drop method. Consequently, the sample was transferred to a solid surface plate (TO-431R00 Polished Steel). To acquire the MS spectra, the following parameters were used: nitrogen LASER 337.1 nm (LTB MLN100), positive ion mode, low mass gate of 400 Da and mass range of 500 to 1500 Da. For MALDI-ToF spectrum calibration, a standard peptide mixture (BSA, ACTH 18–39 for linear mode, Cytochrome C, ACTH 7–38 for reflectron mode) was used. The final mass spectrum was obtained as a result of 200 profiles (with two shots per profile) per each acquisition.

The synthesized peptides were also investigated using tandem mass spectrometry through CID of the protonated structure. The mass range was set from 100 to 600 Da, with 20 shots per profile and 40 profiles per spectra and CID with collision energy of 20 keV with the helium mode. The ion gate was set within 5 Da of the observed mass by ToF/ToF experiments, and the pulsed extraction was optimized for each peptide mass. All values obtained following MS/MS analysis were well-correlated with the theoretical data obtained with GPMAW 6.11 software.

### 4.7. Circular Dichroism

In order to analyze the conformations of the selected peptides, we utilized a theoretical circular dichroism (CD) [72,73]. The CD spectra were recorded in a range of 150 to 350 nm, by monitoring the ellipticity (deg·cm^2^/dmol) with a 1 nm step. This analysis also included backbone charge-transfer transitions of the peptides and peptide-based drugs. The peptides and peptide-based drugs were designed in MOE software as described in Section 2.3, and their structures were further subjected to software developed by Bulheller et al. in order to obtain the unprocessed data and to the calculation provided by Wiedemann et al. to analyze the acquired plots. This methodology was adopted because the in silico analysis provided us the exact CD spectra of our designed peptides, regardless of the nature of the in vitro interference.

## 5. Conclusions

Our study offers a unique insight into the design of neuropeptide-based drugs with β-sheet breaker potential in the amyloid-beta cascade. In our work, we investigated and characterized over 70 peptide-based drugs with the potential ability to break the β-sheet structures of amyloid-beta peptides from senile plaques. During the preliminary analysis, we obtained exciting data, which were applied theoretically—in silico—and experimentally—in vitro.

Furthermore, our approaches demonstrated that the peptides ^17^LVFF^20^, NA-^17^LVFF^20^, ^16^KLVF^19^ and NA-^16^KLVF^19^ have the ability to interfere with the fibril formation and therefore to improve the neuro and cognitive functions. In addition, the peptide NA-^16^KLVF^19^ possesses attractive pharmacological features which will be applied in our further in vitro and in vivo experiments. Hence, all these data could offer relevant approaches regarding innovative therapeutics pathways for Alzheimer’s disease, with the molecular and structural deciphers of these novel neuropeptide-based drugs with β-sheet breaker potential. Further investigations will be performed in order to classify and evaluate the inhibitory activity against amyloid-beta aggregation.

## Figures and Tables

**Figure 1 ijms-23-02857-f001:**
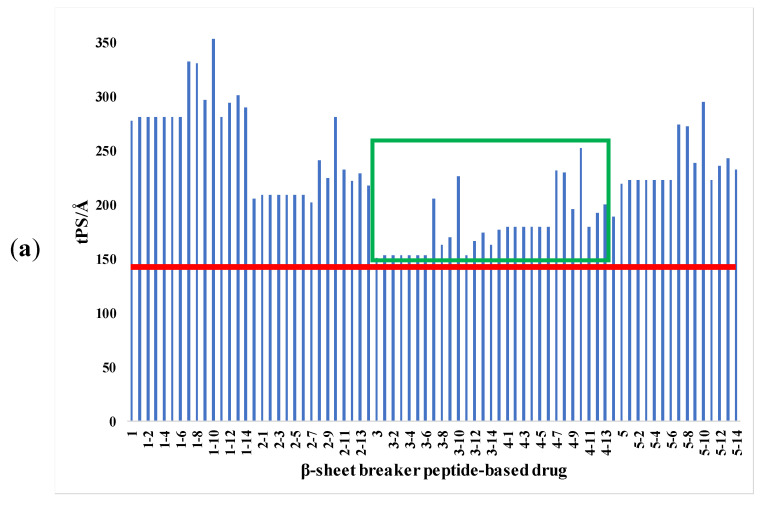
The pharmacological administration properties of the newly designed peptides and derivatives regarding: (**a**) the polarity based on topological surface of the structures, where the red line denotes the limit; (**b**) the lipophilicity of the structures, where the blue lines denote the limits of this parameter.

**Figure 2 ijms-23-02857-f002:**
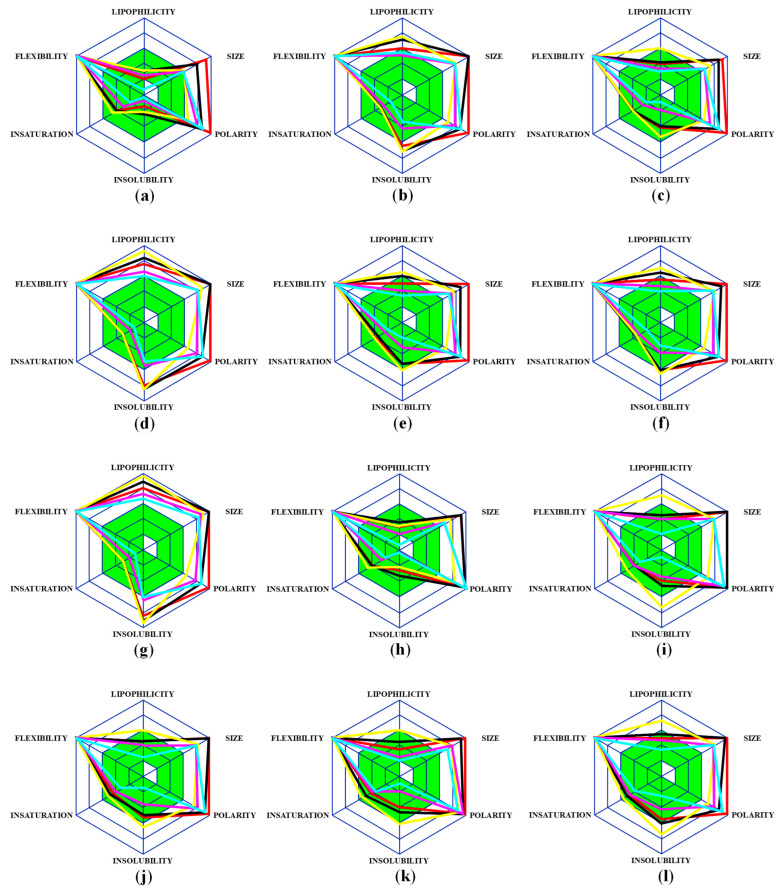
The physicochemical properties for drug delivery based on lipophilicity, size, polarity, insolubility, instauration and flexibility of the peptide: (**a**) 1, 2, 3, 4 and 5; (**b**) 1-1, 2-1, 3-1, 4-1 and 5-1; (**c**) 1-2, 2-2, 3-2, 4-2 and 5-2; (**d**) 1-3, 2-3, 3-3, 4-3 and 5-3; (**e**) 1-4, 2-4, 3-4, 4-4 and 5-4; (**f**) 1-5, 2-5, 3-5, 4-5 and 5-5; (**g**) 1-6, 2-6, 3-6, 4-6 and 5-6; (**h**) 1-7, 2-7, 3-7, 4-7 and 5-7; (**i**) 1-8, 2-8, 3-8, 4-8 and 5-8; (**j**) 1-9, 2-9, 3-9, 4-9 and 5-9; (**k**) 1-10, 2-10, 3-10, 4-10 and 5-10; (**l**) 1-11, 2-11, 3-11, 4-11 and 5-11; (**m**) 1-12, 2-12, 3-12, 4-12 and 5-12; (**n**) 1-13, 2-13, 3-13, 4-13 and 5-13; (**o**) 1-14, 2-14, 3-14, 4-14 and 5-14. Red—peptide 1, blue—peptide 2, yellow—peptide 3, pink—peptide 4 and aqua—peptide 5. The green hexagon denotes the acceptable values.

**Figure 3 ijms-23-02857-f003:**
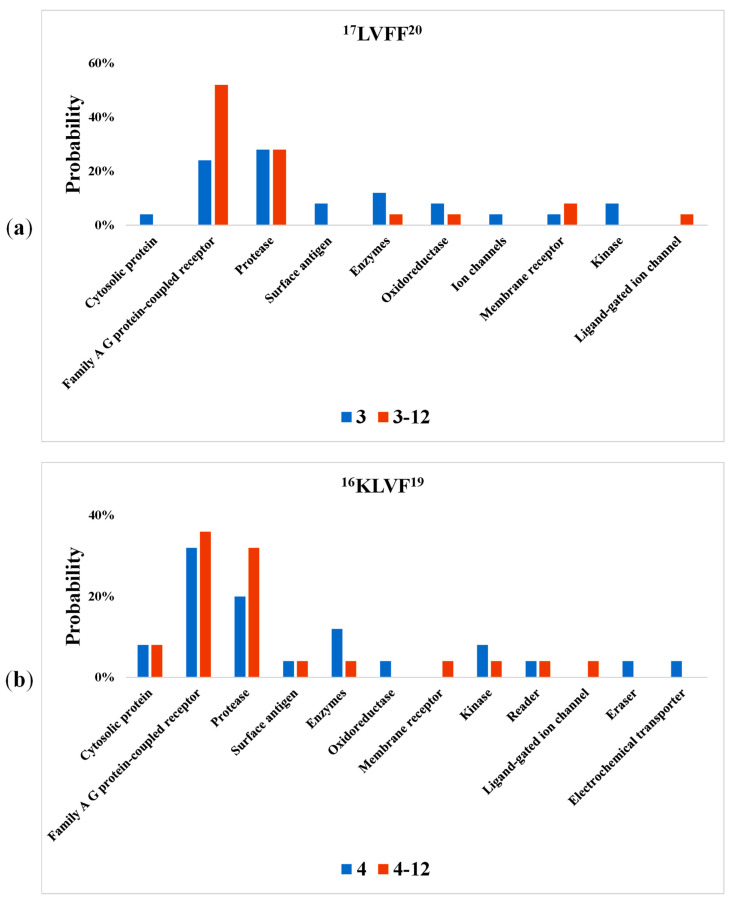
The bioactivity screening regarding the probability of interaction between each class of active biomolecules and peptide–nicotinic acid conjugates with the following sequences: (**a**) ^17^LVFF^20^, (**b**) ^16^KLVF^19^.

**Figure 4 ijms-23-02857-f004:**
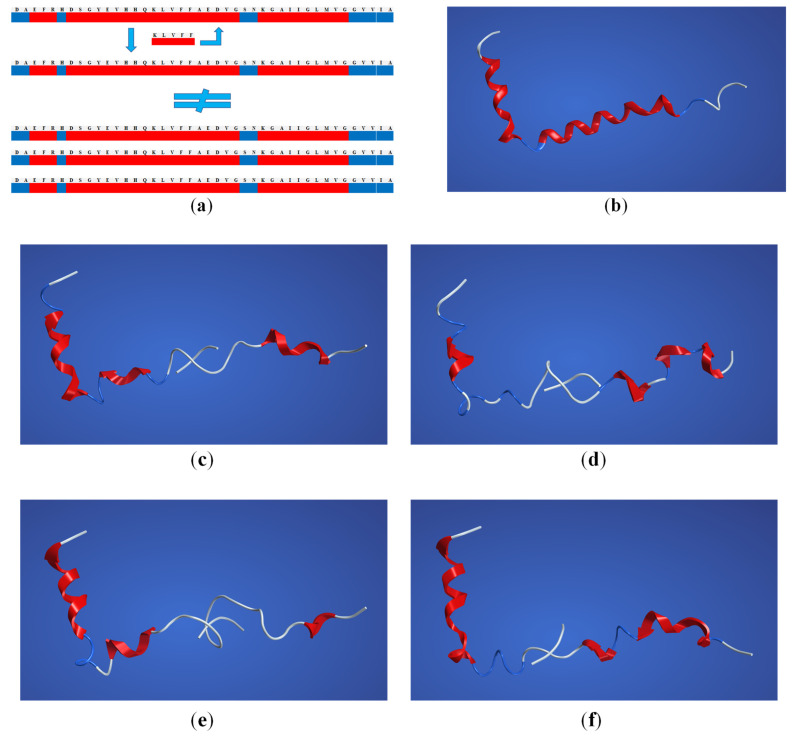
(**a**) The proposed β-sheet breaker mechanism of fibril formation through peptide-based drug binding with Aβ_(1–42)_ monomer. (**b**) The crystallographic structure of Aβ_(1–42)_ monomer obtained from the Protein Data Bank and built with MOE software. The alignment pattern of Aβ_(1–42)_ with (**c**) ^17^LVFF^20^; (**d**) NA-^17^LVFF^20^. The alignment pattern of Aβ_(1–42)_ with (**e**) ^16^KLVF^19^ and (**f**) NA-^16^KLVF^19^.

**Figure 5 ijms-23-02857-f005:**
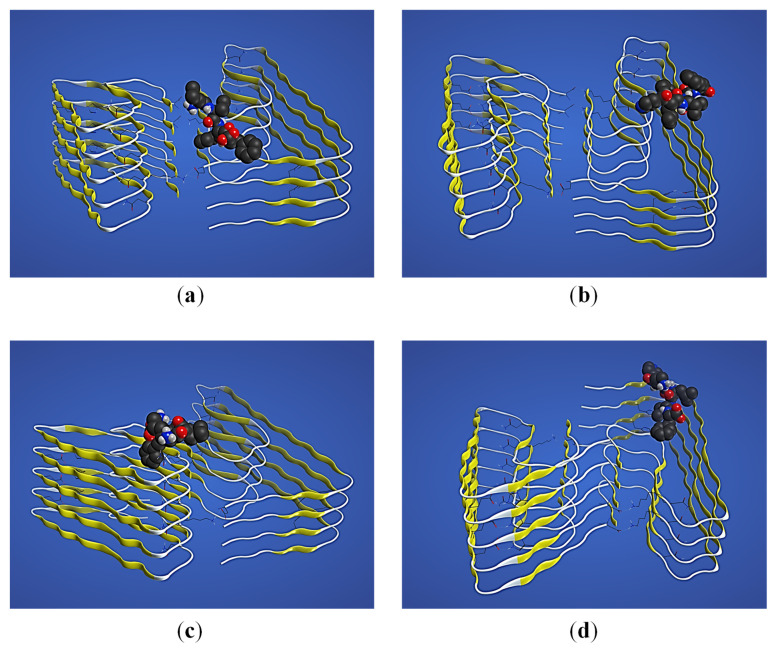
The docking simulation between Aβ_(1–42)_ fibrils (yellow β-sheet structures) and the proposed peptide-based drugs (black structures): (**a**) ^17^LVFF^20^ and (**b**) NA-^17^LVFF^20^; (**c**) ^16^KLVF^19^ and (**d**) NA-^16^KLVF^19^.

**Figure 6 ijms-23-02857-f006:**
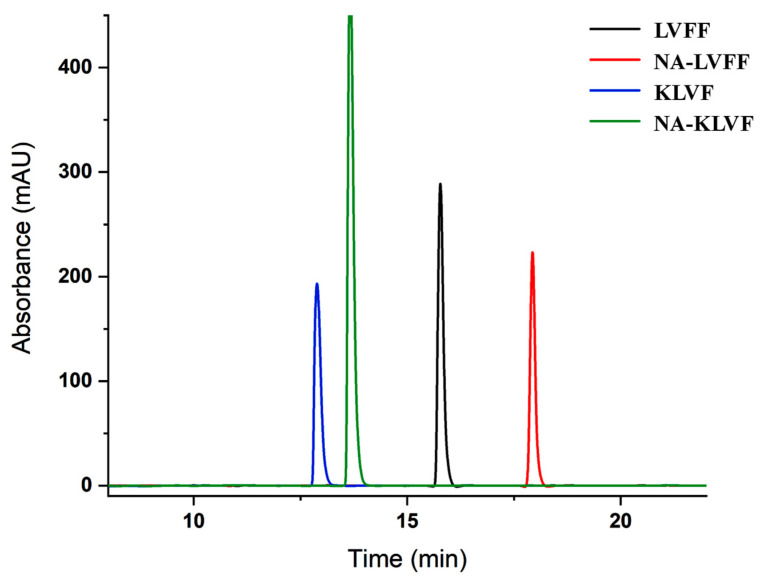
The chromatographic separation of purified peptides by RP-HPLC: ^17^LVFF^20^ (black), NA-^17^LVFF^20^ (red), ^16^KLVF^19^ (blue) and NA-^16^KLVF^19^ (green).

**Figure 7 ijms-23-02857-f007:**
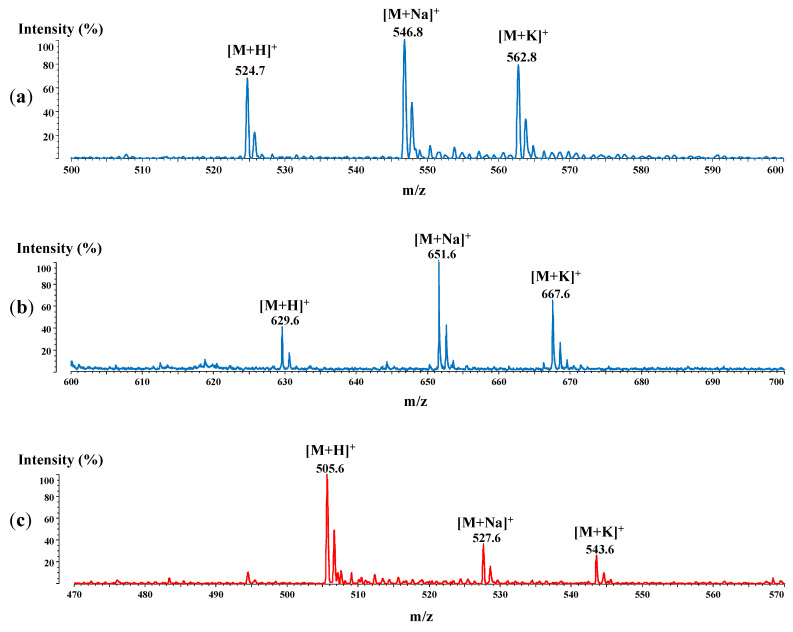
The MALDI-ToF/ToF spectra of the synthesized peptides: (**a**) ^17^LVFF^20^; (**b**) NA-^17^LVFF^20^; (**c**) ^16^KLVF^19^ and (**d**) NA-^16^KLVF^19^.

**Figure 8 ijms-23-02857-f008:**
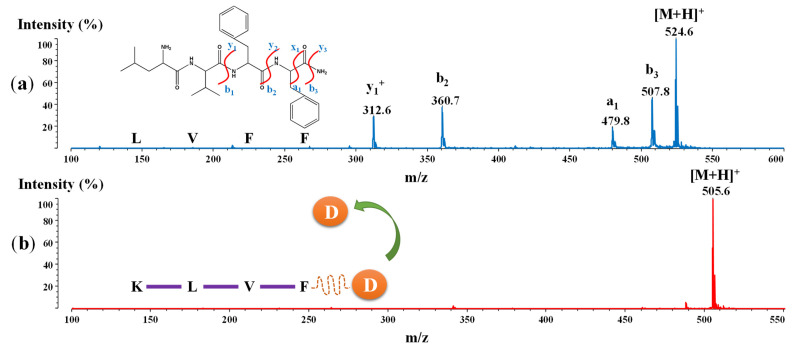
The mass spectra following collision-induced dissociation in positive reflectron mode of: (**a**) ^17^LVFF^20^ and (**b**) ^16^KLVF^19^ peptides, where the orange D represent the additional drug.

**Figure 9 ijms-23-02857-f009:**
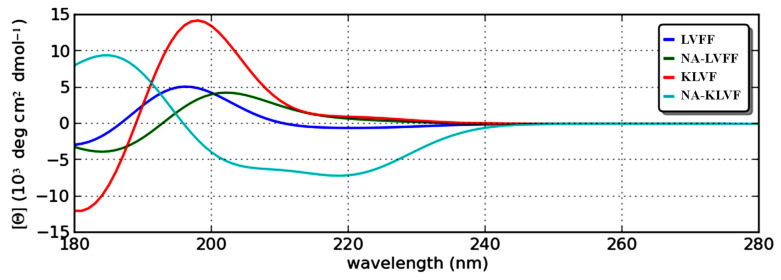
Circular dichroism spectra of ^17^LVFF^20^, NA-^17^LVFF^20^, ^16^KLVF^19^ and NA-^16^KLVF^19^ peptides.

**Table 1 ijms-23-02857-t001:** The design of novel β-sheet breaker drugs based of amyloid-beta (15–20) peptide.

Code		OC	Code		OC	Code		OC	Code		OC	Code		OC
Peptide ↓			Peptide ↓			Peptide ↓			Peptide ↓			Peptide ↓		
**1-1**	1	1	**2-1**	2	1	**3-1**	3	1	**4-1**	4	1	**5-1**	5	1
**1-2**	1	2	**2-2**	2	2	**3-2**	3	2	**4-2**	4	2	**5-2**	5	2
**1-3**	1	3	**2-3**	2	3	**3-3**	3	3	**4-3**	4	3	**5-3**	5	3
**1-4**	1	4	**2-4**	2	4	**3-4**	3	4	**4-4**	4	4	**5-4**	5	4
**1-5**	1	5	**2-5**	2	5	**3-5**	3	5	**4-5**	4	5	**5-5**	5	5
**1-6**	1	6	**2-6**	2	6	**3-6**	3	6	**4-6**	4	6	**5-6**	5	6
**1-7**	1	7	**2-7**	2	7	**3-7**	3	7	**4-7**	4	7	**5-7**	5	7
**1-8**	1	8	**2-8**	2	8	**3-8**	3	8	**4-8**	4	8	**5-8**	5	8
**1-9**	1	9	**2-9**	2	9	**3-9**	3	9	**4-9**	4	9	**5-9**	5	9
**1-10**	1	10	**2-10**	2	10	**3-10**	3	10	**4-10**	4	10	**5-10**	5	10
**1-11**	1	11	**2-11**	2	11	**3-11**	3	11	**4-11**	4	11	**5-11**	5	11
**1-12**	1	12	**2-12**	2	12	**3-12**	3	12	**4-12**	4	12	**5-12**	5	12
**1-13**	1	13	**2-13**	2	13	**3-13**	3	13	**4-13**	4	13	**5-13**	5	13
**1-14**	1	14	**2-14**	2	14	**3-14**	3	14	**4-14**	4	14	**5-14**	5	14

The peptides are ^15^QKLVFF^20^—1, ^16^KLVFF^20^—2, ^17^LVFF^20^—3, ^16^KLVF^19^—4, ^15^QKLV^18^—5 and the organic compounds used for conjugation (OC) are lauric acid—1, hexanoic acid—2, palmitoyl acid—3, octanoic acid—4, decanoic acid—5, stearoyl acid—6, taurine—7, 4-hydroxy-3-methoxymandelic acid—8, indole-3-butyric acid—9, 3,5-dinitrosalicylic acid—10, naphthaleneacetic acid—11, nicotinic acid—12, salicylic acid—13 and acetylsalicylic acid—14.

## Data Availability

Not applicable.

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
