# Peer review of "Novel Design of Neuropeptide-Based Drugs with β-Sheet Breaking Potential in Amyloid-Beta Cascade: Molecular and Structural Deciphers"

_ijms, 2022, doi:10.3390/ijms23052857_

Round 1
Reviewer 1 Report
The manuscript is well written and the data presented is also good.

Reviewer 2 Report
Herein, the authors present a study of combinations between neuropeptide drugs with β-sheet and various organic acids and their properties though analytical and computational approaches. The biocompatibility investigation as well as the structural interactions among the derivatives is quite interesting. In general, the provided information is satisfactory and the manuscript is well written. I have some comments and suggestions:
The authors should stress more the selection of different organic acid. They could highlight, also in the abstract, the importance of these sequences and the biological correlation according to their physicochemical properties.
What does it mean ‘’Target prediction for Homo sapiens’’ (section 2.2)? Target prediction and bioactivity screening can be used.
Regarding the structural model and the constructs patterns, can the authors inform about the environment of the target (such as polarity, conformation) and stress more any possible interactions with the amino and carboxyl groups?
